# Inspiratory Muscle Training Improves Aerobic Fitness in Active Children

**DOI:** 10.3390/ijerph192214722

**Published:** 2022-11-09

**Authors:** Ching-Hsin Lin, Chih-Wei Lee, Chien-Hui Huang

**Affiliations:** 1Division of Rehabilitation Technology, Hualien Tzu Chi Hospital, Buddhist Tzu Chi Medical Foundation, Hualien 970473, Taiwan; 2Department of Physical Therapy, Tzu Chi University, Hualien 970374, Taiwan

**Keywords:** maximal inspiratory pressure, six-minute walk distance, aerobic fitness, VO_2_max, inspiratory muscle training

## Abstract

Research on the effect of inspiratory muscle training (IMT) on exercise performance is inconsistent. IMT has not been applied to fit child athletes, who are more likely to develop inspiratory muscle fatigue, and, consequently, to benefit from IMT. Methods: Thirty children (mean age: 10.7 ± 1.2 years) were recruited and randomly assigned to a high-intensity (HG), a low-intensity (LG), or a control group (CG). For both training groups, a double-blind procedure was applied. In the HG, 80% of maximal inspiratory pressure (MIP) was used as the level of training intensity. The LG used 30% MIP. Training groups were trained at 6 breaths a set, 4 sets a day, totaling 24 breaths a day for 6 weeks. Exercise capacity outcomes include maximal and submaximal aerobic capacity, as measured as VO_2_max and distance from six-minute walk test (6MWD). Results show improvement in MIP, VO_2_max, and 6MWD only in the HG. MIP in the HG significantly increases from 108.7 (100.8–143.3) to 144.4 (130.0–175.6) cmH_2_O. VO_2_max in the HG increases from 43.0 (40.5–45) to 53.0 (46–63) mL·kg^−1^·min^−1^. The 6MWD increases from 792.0 (737.5–818.0) to 862.0 (798.5–953.5) m. Data are presented as median (interquartile range). No difference is found in the LG or CG. Conclusion: high-intensity IMT increases MIP, maximal, and submaximal exercise capacity in the HG, but no difference is found in the LG or CG. Therefore, high-intensity type of IMT improves aerobic fitness in fit children by appropriately applying inspiratory muscle strength training.

## 1. Introduction

Ventilation systems are not considered a limiting factor to maximal exercise at sea level because arterial oxygen pressure is generally well-maintained, even during high-level exercise [1]. However, most relevant studies demonstrate that the diaphragm fatigues following high-level exercise [2]. Athletes who have to train at a higher intensity for longer period of time are likely to suffer from diaphragm fatigue. Research shows that diaphragm fatigue induces sympathetically mediated reductions in limb blood flow and can further diminish endurance performance [3].

Inspiratory muscle training has been applied to different populations, ranging from low-fitness groups, such as patients with cardiopulmonary disorders, to high-fitness groups, such as elite athletes. The purpose of inspiratory muscle training is mostly to increase exercise performance, but the effects are inconsistent due to the variant outcomes measured, the task required, and subjects from different levels of breathing fitness. Respiratory muscle fatigue is found during high-intensity exercise (>85% VO_2_max) and would further limit exercise performance by increased sympathetic vasoconstrictor outflow to limb muscles and the sensation of dyspnea [4]. Diaphragm fatigue is considered present if there is a ≥15% reduction in maximal pressure ability, and is a result of the combination of pressure generating during inspiration and the ratio of inspiration time to total breathing cycle duration [5]. The rationale for a strength type of inspiratory muscle training (IMT) is based on ventilatory demand requiring higher pressure-generating capability during incremental to maximal exercise to produce increased volume and airflow. The strength type of IMT was specifically developed to increase inspiratory strength, and was found to increase diaphragm thickness [6] and aerobic performance [7], as well as relieving the sensation of dyspnea [8]. However, data on the effects of IMT on exercise performance, especially for athletes, are equivocal. The difference in outcomes probably depends on how well the principles of training are applied, the specific type of training, and the nature of subjects that receive the training.

Children have smaller lungs than adults, and women have smaller lungs and airways compared to men. Research on breathing in children is scarce, but evidence shows that work of breathing in females increases more rapidly than males and females present a greater O_2_ cost of breathing when matched for a relative amount of ventilation [9]. For the same reason, children can suffer more from inspiratory muscle fatigue during high intensity exercise.

Nowadays, young athletes are achieving higher performances in many sports and they probably respond differently to training than adults. If IMT is necessary to prevent or delay inspiratory muscle fatigue, children and athletes would benefit from it most because of their smaller lungs and having to exercise at high intensity, respectively. IMT can prevent or delay fatigue, which provides an athlete with a competitive advantage. Our review of the literature of respiratory muscle training and exercise performance revealed that no single study coupled IMT with maximal aerobic capacity exercise performance for fit child athletes. Therefore, the purpose of this study was to investigate the effect of a strength type of IMT on maximal oxygen uptake (VO_2_max) in extremely fit child athletes.

## 2. Materials and Methods

The study was designed to evaluate the effect of a 6 week IMT program on inspiratory muscle strength and exercise performance in fit young boys. To explore the difference in training effects between high- and low-intensity types of IMT, random assignment and double-blind control design were applied. All of the participants were randomly assigned to a high-intensity training group (HG), a low-intensity training group (LG), or a no-training control group (CG). The HG and LG each received 6 weeks of IMT. The CG did not receive any inspiratory training. For both training groups, double-blind control was applied so that the group decision was unknown to the participants and experiment operator. Subjects in both training groups received a personal inspiratory trainer with pressure threshold set at a pre-determined intensity. The intensity setting on the inspiratory trainer was occluded with tape and the intensity was not seen by the subjects.

### 2.1. Participants

Boys from local elementary sports teams were recruited. The inclusion criteria were as follows: (1) at least 2 years of training experience with the team; and (2) the ongoing minimal sports training amount that the participants received was 90 min/d, for at least 5 days a week.

Thirty regularly trained, non-smoking boy volunteers with normal lung function were studied (FEV1 > 80% of age predicted, FVC > 80% of age predicted, and FEV1/FVC > 70%) [10]. The study was approved by the Institutional Review Board in Tzu Chi Medical Center. Informed consent was obtained from each subject and their guardian.

All subjects assessed for eligibility met the inclusion criteria of the study. The inclusion criteria of a two-year training background and sufficient ongoing training were confirmed by the coaches. Thirty children were randomized before initiating the study protocol. A prior study evaluated the effect of IMT on MIP for healthy subjects in our laboratory, and demonstrated the significant difference of the mean MIP of 20 cmH_2_O [11]. Based on this difference, and the study objective to realize 80% power with type I error of 0.05, the present study required twenty-one children for the three study groups when using analysis of variance (ANOVA). Hence, the study included thirty boys to account for possible a 30% dropout.

Weight and height were measured, and the body mass index (BMI) was calculated. Body composition was assessed using bioelectrical impedance (Biospace Inbody 3.0 Body Composition Analyzer, Singapore) to determine body fat percentage.

### 2.2. Outcome Measurements-Inspiratory Muscle Strength

Maximal inspiratory pressure (MIP) was measured at the mouth by a pressure manometer (RPM, Micro, Cardinal Health, Basingstoke, UK). Subjects were seated in a comfortable chair upright with nose clips on. After exhaling to residual volume, subjects placed their lips around a mouthpiece and inspired as forcefully as possible. Repeated measurements were taken at least five times, with a 60 to 120 s rest between trials, until three measurements within 10% variation were obtained. The average of these three measurements was recorded as the subjects’ MIP. MIP is the highest pressure that subjects can generate and represented inspiratory strength.

### 2.3. Outcome Measurements-Maximal Exercise Capacity

Maximal exercise capacity was assessed using maximal oxygen uptake (VO_2_max) by having the participants run on an electromechanical treadmill (HP-Cosmos, Quasar Med 4.0, Nussdorf–Traunstein, Germany). The treadmill was programmed for increases in angle of inclination and speed according to the original Bruce protocol. The boys were instructed to avoid vigorous exercise within 2 h of the beginning the test, and not to hold the guardrail during the test. Breath-by-breath gas samples were collected using a fitted facemask and were analyzed throughout the test using open-circuit calorimetry. A two-way breathing valve was attached to the mask, which covered their nose and mouth, and was used to collect the expired air. The air was analyzed continuously for ventilatory and metabolic variables. During the maximal exercise test, physiologic responses were measured using a heart rate (HR) monitor (Polar, Polar Electro, Finland) and a calibrated mobile gas analysis system (Cortex Metamax 3B, Cortex Medical GmbH, Leipzig, Germany). The Cortex Metamax 3B is a valid and reliable system for measuring ventilatory parameters during exercise [12]. VO_2_max was determined when 2 of the following 3 conditions were reached: (1) respiratory exchange ratio > 1.0; (2) heart rate > 85% of the age-predicted maximum; and (3) the child was exhausted and refused to continue, despite strong verbal encouragement [13]. VO_2_max was defined as the highest value achieved during the last 30 s before peak or levelling. During the test all subjects were highly motivated and strongly encouraged to continue to run to the maximum workload.

### 2.4. Outcome Measurements-Submaximal Exercise Capacity

Submaximal exercise capacity was assessed by the six-minute walk test (6MWT), which was conducted according to guidelines recommended in an official ATS statement [14]. Dyspnea, as measured with the visual analogue scale (VAS), peripheral oxygen saturation (SpO_2_), and pulse rate were measured using pulse oximetry at the start and end of the 6MWT. VAS was measured using a horizontal line, 100 mm in length, anchored by the word descriptors “no dyspnea at all” and “very severe dyspnea” at each end. Subjects were asked to mark on the line the point that represented their perception of their state. The VAS score was determined by measuring in millimeters from the left-hand end of the line to the point that the patient marked. The distance that the subjects covered in 6 min was recorded as the 6MWD. The 6MWD is a functional measurement and represented a submaximal aerobic exercise capacity.

### 2.5. Inspiratory Muscle Training (IMT) Protocol

Training for both the HG and LG was conducted 5 days per week for 6 weeks. Each daily inspiratory training session consisted of 4 sets of 6 training breaths. Each participant used a personal custom-designed pressure threshold inspiratory muscle trainer, which was a device that provided a pressure threshold load to inspiration so that the participant must generate pressure greater than the load to open the valve for air to enter. Pressure threshold inspiratory trainer with individualized inspiratory training load was pre-set according to the group decision before giving it to the subjects. In the HG, training intensity was 80% of MIP, whereas in the LG, training intensity was 30%. The two training groups performed inspiratory training daily at the beginning of their daily routine sports team practice, completing 4 sets of training required for approximately 10 min. During the first week, the same researcher in this study supervised the training to ensure that there was no difficulty in learning or practicing. After the participants successfully learned the skill to breath against the pressure-loaded trainer, the team coach assumed the supervision. At the end of the third week and in the remaining weeks, every week the researcher remeasured the participants’ MIP and adjusted the pressure threshold to 80% or 30% of the current MIP.

### 2.6. Statistical Analysis

Our subjects were active children of a similar age and the sample size was small (n = 30). Not all data in this study passed the Shapiro–Wilk test for normality assumptions, therefore, non-parametric statistics were adopted for the following analysis. All data are presented as the median (interquartile range), unless otherwise explained. Anthropometric data (body mass, height, weight, body mass index (BMI), and body fat percentage), inspiratory muscle strength data (MIP), maximal exercise capacity (VO_2_max), and submaximal exercise capacity (6MWD) were compared using Kruskal–Wallis one-way analysis of variance (ANOVA) on ranks. When significant differences are found, Dunn’s test was used for pairwise comparison. Within-group change between pre- and post-IMT was evaluated using the Wilcoxon signed-rank test. Significance levels for all tests were established as two-tailed *p* values less than 0.05.

## 3. Results

### Study Group Characteristics

Thirty boys were randomly assigned into three groups (HG: n = 9, mean ± SD age 12.1 ± 0.3 y, height 149.2 ± 9.3 cm, weight 44.1 ± 9.5 kg; LG: n = 10, age 10.3 ± 1.3 y, height 140.4 ± 12.5 cm, weight 34.9 ± 9.7 kg; and CG: n = 11, age 10.0 ± 0.6 y, height 138.9 ± 8.2 cm, weight 36.4 ± 9.1 kg). The participants completed 6 weeks of inspiratory muscle training (from both training groups) and all participants finished both pre- and post-training measurements. All participants were highly motivated and maintained their original daily sports training routine during our experiment period. The sports training was twice a day, one hour before and one hour after school. Subjects were preparing for a national middle school contest when the experiment was conducted. The randomization procedure results in significant differences in age among groups, and the following Dunn’s test shows that the age in the HG is significantly greater than both the LG and CG (Table 1). For height and weight, differences are found in the Kruskal–Wallis ANOVA test, but the following Dunn’s test shows that there is no difference in any pairwise comparison. There are no differences in other physiological characteristics such as BMI or body fat percentage.

For the pre-IMT outcome variables, results from the Kruskal–Wallis test show that there is significant group difference in VO_2_max and 6MWD (*p* < 0.05). A post hoc Dunn’s test was used for pairwise comparison. VO_2_max is significantly greater in the LG when compared to the HG. The 6MWD is found to be smaller in the CG when compared to both the HG and LG (Table 2).

For the within group post-IMT outcome variables, results from the Wilcoxon signed-rank test show significant improvement in MIP, VO_2_max, and 6MWD in the HG (*p* < 0.05). MIP, VO_2_max, and 6MWD do not change in the LG or CG. The detailed outcome data are presented in Table 2. MIP in the HG increases from 108.7 pre-training (100.8–143.3) to 144.4 (130.3–175.6) cmH_2_O post-training (*p* < 0.05). MIP in the LG and CG does not change, ranging from 114.4 (83.8–138.5) and 137.0 (95.0–158.0) pre-training to 109.7 (96.3–135.7) and 153.0 (105.0–207.0) cmH_2_O post-training, respectively, as shown in Figure 1. This suggests that 6 weeks of high-intensity IMT results in an increase in MIP, and low-intensity IMT does not result in any change in MIP.

VO_2_max, expressed as mL·kg^−1^·min^−1^, increases only in the HG after 6 weeks of training and remains unchanged in both the LG and CG. The Wilcoxon signed-rank test shows that the HG significantly improves VO_2_max, which in the HG ranges from 43.0 (40.5–45.0) pre-training to 53.0 (46.0–63.0) mL·kg^−1^·min^−1^ post-training (*p* < 0.05). VO_2_max in the LG and CG do not change, ranging from 49.5 (45.7–55.3) pre-training to 54.5 (49.7–58.3) mL·kg^−1^·min^−1^ post-training in the LG, and 49.0 (45.0–52.0) pre-training to 46.0 (42.8–50.3) mL·kg^−1^·min^−1^ post-training in the CG, as shown in Figure 2.

Wilcoxon signed-rank test shows that the 6MWD improves significantly in the HG. The 6MWD in the HG improves from 792.0 (737.5–818.0) pre-training to 862.0 (798.5–953.5) m post-training (*p* < 0.05). However, the 6MWD in the LG and CG remain unchanged, which range from 834.5 (797.5–900.8) pre-training to 847.5 (812.8–890.5) m post-training, and from 726.0 (626.0–750.0) pre-training to 713.0 (688.0–751.0) m post-training, respectively, as shown in Figure 3. There is no difference in HR, SpO_2,_ and VAS after training among groups during the 6MWT (Table 3).

The results suggest high-intensity IMT improves inspiratory muscle strength, maximal, and submaximal aerobic exercise capacity, as seen in the MIP, VO_2_max, and 6MWD values, respectively. In contrast, low-intensity IMT does not show any effects on MIP, VO_2_max, or 6MWD.

## 4. Discussion

This study aimed at evaluating the impacts of IMT on inspiratory strength, maximal, and submaximal exercise capacity, hypothesizing that high-intensity, but not low-intensity, IMT could produce improvement. Random assignment and double-blind control design were applied to compare the effect. The findings verify our hypotheses and show that high-intensity IMT improves inspiratory strength, and maximal and submaximal exercise capacity among physically fit child athletes. Low-intensity IMT does not improve inspiratory strength, and no difference in exercise capacity is found.

### 4.1. Effects of Inspiratory Muscle Training on MIP

Maximal inspiratory pressure is used for the assessment of inspiratory muscle strength. However, one difficulty is that this test requires subjects to understand and follow the instructions and to perform with maximum effort. The normative data for children are limited, and for youth athletes are unavailable. The MIP values found in the current study are greater than children of a similar age [15]. The possible reason could be that our subjects are physically fit child athletes. Tests of MIP require understanding, cooperation, and patience, and we carefully instructed and monitored the performance. The requirement was that each subject had to perform the tests multiple times with enough rest in between and the highest three scores of tests had to be within 10% variation. This was to avoid inconsistent high outlier data due to accidental biting of the mouthpiece. We have not found studies that compare the MIP between healthy subjects and athletes at this young age.

Our high intensity IMT results in an average of a 35% increase in MIP in the HG; low-intensity training does not affect MIP. The training outcome confirms training specificity theory, because inspiratory muscles are morphologically and functionally skeletal muscles and would respond to appropriate physiological stimuli. Training received in the HG is a model of strength training, which is high intensity (80% of MIP), low repetition (six breaths a set), aiming at an increase in maximal strength (MIP). The amount of training effect is similar to reports on healthy adult athletes [16], suggesting that children at this age respond to inspiratory training stimuli as effectively as adults. This positive finding is also consistent with previous studies about weight training for youths [17].

### 4.2. Effects of Inspiratory Muscle Training on Maximal Exercise Capacity VO_2_max

Normally trained young athletes have higher VO_2_max than their untrained peers, and VO_2_max increases in boys through childhood and adolescence; VO_2_max values of >60 mL/kg/min have been observed [18]. Compared to the above review, our participants (mean age 10.7 y; median age 10.4 y) are relatively younger and present an average pre-training VO_2_max of 48.1 mL/kg/min. However, if compared to the data from the literature on VO_2_max in Chinese boys [19], which show the average VO_2_max of 30 mL/kg/min from boys aged 8–12 years, we believe that VO_2_max of 48.1 mL/kg/min pre-training from our participants is able to represent a group of trained boy athletes appropriate to their age.

The finding in which an increase in VO_2_max improves after IMT in the HG suggests an improvement in maximal aerobic exercise capacity. The VO_2_max in the HG is 44.9 ± 8.2 (mean ± SD) pre-IMT and 53.4 ± 9.5 post-IMT, which represents a 21% increase. There is considerable evidence that inspiratory muscles develop fatigue during intense exercise, considering the significant demand for O_2_ and the high work of breathing that occurs during heavy endurance exercise [20]. It has been established that respiratory muscle fatigue occurs during exercise at an intensity of at least 85% of maximal oxygen consumption [21]. The outcome of inspiratory muscle fatigue is that fatigue causes a sympathetically mediated vasoconstriction of the lower limbs and further results in peripheral muscle fatigue, known as inspiratory muscle metaboreflex [22]. The mechanism involves neural and cardiovascular change, which impairs exercise performance. IMT induces the delay in activating this mechanism, decreasing the onset of peripheral muscle fatigue [20] and the sensation of dyspnea [11]. The improvement in our study is in agreement with a previous finding by Holm et al. [23], which demonstrates that IMT improves VO_2_max in a group of fit cyclists. Training builds stronger inspiratory muscles, which helps delay the onset of inspiratory muscle fatigue; the possible sympathetic reflex, which may lead to reduced blood flow in the peripheral muscle, is, thus, prevented. Hajghabari et al. [24] published a systemic review of the effects of respiratory muscle training on the performance of athletes and concluded that respiratory muscle training improved performance. In their conclusion, the authors offered the important explanation for inconsistent outcomes in IMT studies: not enough progression in training intensity. The present training protocol uses 80% of MIP as the training intensity, 24 breaths per day over 6 weeks of training, and with weekly adjustment for training intensity. To delay possible inspiratory muscle fatigue, and to prevent further metabolic reflex, it is necessary to apply overload and specificity training principles for improving strength.

### 4.3. Effects of Inspiratory Muscle Training on Submaximal Exercise Capacity

The 6MWD test is a reliable and valid functional test for assessing exercise tolerance and endurance in healthy children [13]. We chose it because it represents the most suitable and convenient method for assessing the submaximal level of functional exercise capacity. Our results show that IMT improves the 6MWD in the HG, without a significant difference in the change of VAS or SpO_2_, meaning that our participants in the HG are able to walk longer distances with the same level of breathlessness and oxygen saturation. Our 6WMD is greater than published data on “healthy children” with comparable age [25], confirming the greater fitness of our participants. After IMT, the 6MWD even improves 10% (786 ± 63 m pre-training to 863 ± 97 m post-training, data presented as mean ± SD) in the HG, while there is no change in the LG (847 ± 63 m pre-training to 852 ± 63 m post-training, data presented as mean ± SD).

### 4.4. Limitations

The first limitation of this study is an insufficient number of subjects, which causes several variables to not pass the Shapiro–Wilk normality Test. We chose to run non-parametric statistical analysis instead for all the dependent variables. It is found that parametric tests are more powerful than non-parametric tests only if all of the assumptions underlying the parametric test are met [26]. Not passing the normality test and small sample sizes both are appropriate conditions in which to use a non-parametric analysis [27]. Second, randomization with small sample results in difference in age among groups. Therefore, the influence of development needs to be considered. Post-puberty boys show greater maximal isometric force than pre-puberty boys [28], and we recognize that this may have influenced our results. Third, we do not differentiate the type of sport team among participants. For a study with small subject number, background endurance type sports, i.e., running, or intermittent type sports, i.e., tennis, could possibly lead to different responses to IMT and affect our results of exercise performance. Thus, larger studies on subgroups representing children with certain sports backgrounds are needed.

### 4.5. Practical Applications

Alongside regular athletic training, researchers and coaches can use additional high-intensity IMT to improve VO_2_max for already fit athletes. This research shows that a strength-type of inspiratory muscle training results in significant increases in both maximal and submaximal exercise capacity. However, an endurance-type of inspiratory muscle training does not have the same effects.

## 5. Conclusions

This study demonstrates that in addition to regular sports training, additional high-intensity IMT improves inspiratory muscle function, accompanied by enhanced exercise capacity, in fit child athletes. At ten minutes per day, this protocol is feasible and efficient. This study is the first to provide evidence that child athletes can respond to IMT. Child athletes benefit from high-intensity IMT. Maximal inspiratory muscle pressure and both maximal and submaximal exercise capacity improve in response to high-intensity IMT. However, low-intensity IMT does not cause changes in MIP or exercise capacity.

## Figures and Tables

**Figure 1 ijerph-19-14722-f001:**
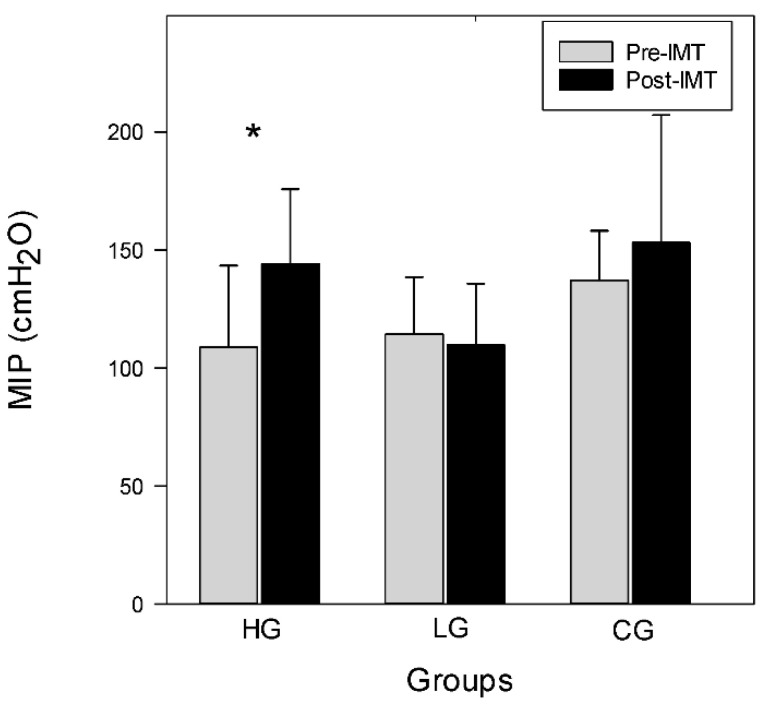
Pre- and post-training MIP among groups. Data are shown as median+ 75th percentile. * *p* < 0.05.

**Figure 2 ijerph-19-14722-f002:**
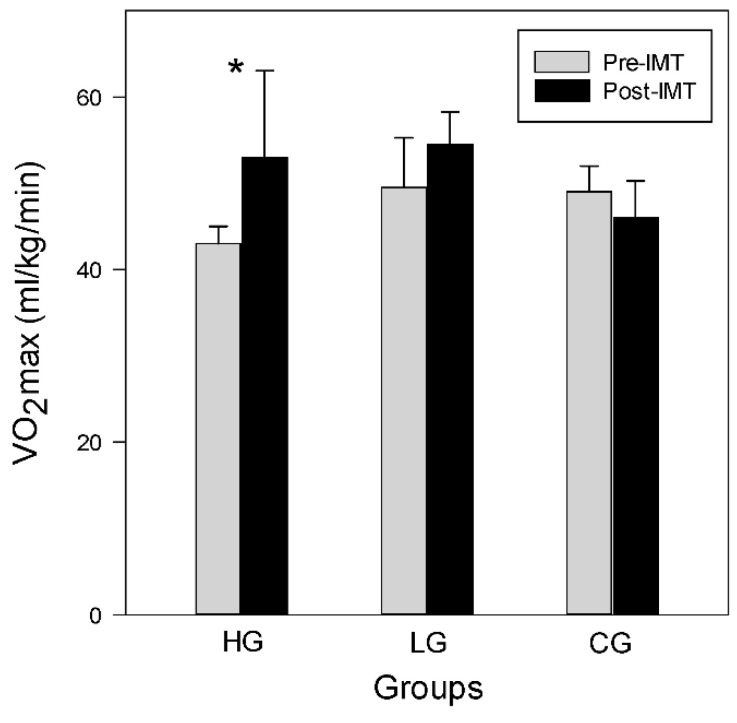
Pre- and post-training VO_2_max among groups. Data are shown as median+ 75th percentile. * *p* < 0.05.

**Figure 3 ijerph-19-14722-f003:**
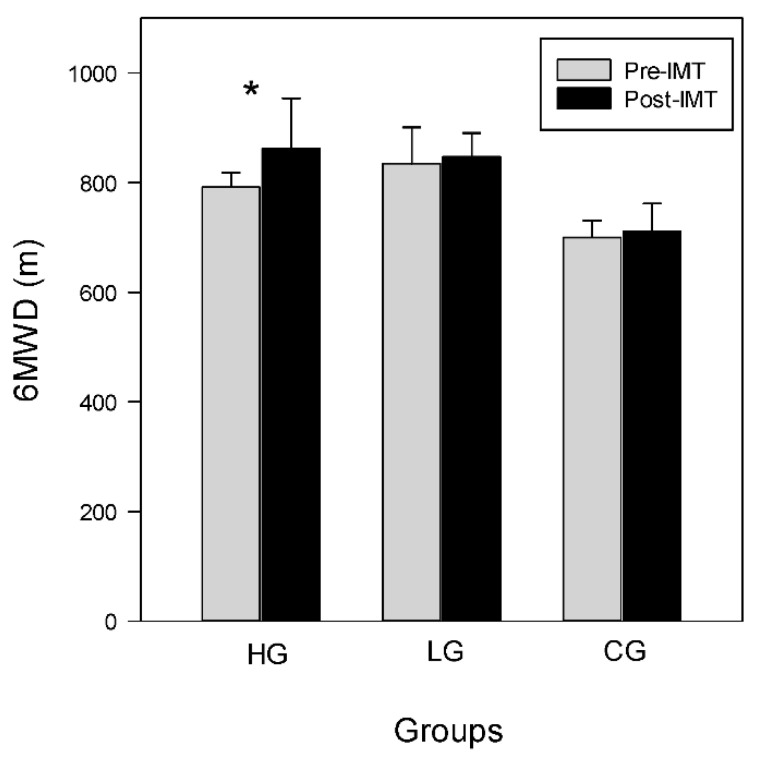
Pre- and post-training 6MWD among groups. Data are shown as median+ 75th percentile. * *p* < 0.05.

**Table 1 ijerph-19-14722-t001:** Comparisons of demographics and anthropometric characteristic among groups.

	Total	HG	LG	CG	*p*-Value
N	30	9	10	11	
Age (years)	10.4 (9.7–12.1)	12.3 (11.7–12.2)	9.7 (9.5–11.6) *	9.8 (9.6–10.4) *	*p* < 0.05
Height (cm)	141.1 (135.4–150.0)	149.0 (141.1–153.9)	137.9 (130.5–147.9)	137.8 (132.3–147.5)	*p* < 0.05
Weight (kg)	35.0 (30.2–44.6)	43.2 (36.4–51.4)	32.1 (29.3–39.2)	30.9 (29.7–42.6)	*p* < 0.05
BMI (kg/m^2^)	18.3 (16.4–20.4)	19.4 (17.7–21.4)	16.9 (15.7–18.8)	17.3 (16.2–20.9)	0.19
Body fat (%)	16.7 (13.1–22.0)	16.7 (13.6–23.7)	14.4 (11.5–20.2)	17.5 (14.7–26.3)	0.42

Data are presented as median (interquartile range). BMI, body mass index. * *p* < 0.05 when compared to HG.

**Table 2 ijerph-19-14722-t002:** Inspiratory strength, maximal exercise capacity, and submaximal exercise capacity before and after IMT among groups.

Dependent Variables	HG	LG	CG	Between Group *p* Value
MIP, cmH_2_O				
Pre	108.7 (100.8–143.3)	114.3 (83.8–138.5)	137.0 (95.0–158.0)	*p* = 0.49
Post	144.4 (130.0–175.6)	109.7 (96.3–135.7)	153.0 (105.0–207.0)	
Within group *p* value	*p* < 0.05	*p* = 0.19	*p* = 0.88	
VO_2_max, mL/kg/min				
Pre	43.0 (40.5–45.0)	49.5 (45.8–55.3) *	49.0 (45.0–52.0)	*p* < 0.05
Post	53.0 (46.0–63.0)	54.5 (49.7–58.3)	46.0 (42.8–50.3)	
Within group *p* value	*p* < 0.05	*p* = 0.19	*p* = 0.43	
6 MWD, m				
Pre	792.0 (737.5–818.0)	834.5 (797.5–900.8)	726.0 (626.0–750.0) *^#^	*p* < 0.01
Post	862.0 (798.5–953.5)	847.5 (812.8–890.5)	713.0(688.0–751.0)	
Within group *p* value	*p* < 0.05	*p* = 0.79	*p* = 0.76	

Data are presented as median (interquartile range). *: *p* < 0.05 when compared to HG. ^#^: *p*< 0.05 when compared to LG.

**Table 3 ijerph-19-14722-t003:** Difference in HR, SpO_2,_ and VAS during 6MWT before and after IMT among groups.

Variables/Groups	HG			LG			CG		
	Beginning of 6MWT	End of 6MWT	Difference	Beginning of 6MWT	End of 6MWT	Difference	Beginning of 6MWT	End of 6MWT	Difference
Pre-IMT									
HR (beats)	96 (84–100)	134 (126–164)	50 (35–64)	89 (84–94)	151 (134–158)	56 (47–72)	98 (90–113)	123 (109–154)	29 (24–57)
SpO_2_ (%)	98 (98–99)	98 (98–99)	−1 (−1–1)	99 (98–99)	98 (98–98)	0 (−1–0)	98 (98–100)	98 (98–98)	0 (−1–1)
VAS (mm)	0 (0–0)	20 (15–35)	20 (15–35)	0 (0–0)	30 (18–45)	30 (15–30)	0 (0–0)	40 (30–50)	30 (23–50)
Post- IMT									
HR (beats)	93 (90–97)	146 (136–176)	74 (46–84)	82 (71–91)	146 (137–166)	69 (61–85)	107 (105–115)	153 (143–173)	54 (37–57)
SpO_2_ (%)	98 (98–98)	98 (97–98)	0 (−1–1)	98 (98–99)	99 (97–99)	0 (−1–2)	98 (98–98)	98 (97–99)	0 (−1–1)
VAS (mm)	0 (0–0)	35 (15–40)	35 (15–40)	0 (0–1.25)	40 (21–40)	35 (20–40)	0 (0–0)	40 (26–41)	40 (25–43)

Data are presented as median (interquartile range).

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
