# Peer review of "Inspiratory Muscle Training Improves Aerobic Fitness in Active Children"

_ijerph, 2022, doi:10.3390/ijerph192214722_

Round 1
Reviewer 1 Report
Thank you for the opportunity to review this manuscript, it is very interesting manuscript with an intriguing topic.
This is an excellent article, well written and complimenting the existing evidence base. I only a minor comment:
Line 81: Could you clarify why boys only were participated in the study?
Apologies if this sounds a bit picky but the rest of the article is so good, I felt this was the only area worth commenting on for clarifying.
Reviewer 2 Report
The scientific work evaluated has a growing interest in the field of rehabilitation. However, I consider that the work does not have enough figures or images and has to be implemented.
Reviewer 3 Report
This article addresses how inspiratory muscle training improves aerobic fitness in children to explore the difference between training intensity. Thus, the author found a difference in the HG and conclude that IMT improves aerobic fitness through appropriate muscle strength training. However, the novel aspects of the manuscript such as the IMT improving aerobic fitness in highly intense trained child athletes were too preliminary and how this is this data is attained to fit athletes with 2 years of prior training experience and ongoing sports training.
Other concerns:
The number of participants is too small, and the data doesn’t represent the total population. Although this was indicated in the limitations.
The age in the experimental group is 10 to 12, but how does the author indicate non-smoke, does he try to refer to non-secondhand smoke? line 86
If the participants were experience trained for 2 years+, how the double-blind design was incorporate into the study? Do the athletes know the exercise that they performed before data collection? Does the author take or observe any other independent measure in the evaluated groups? How does the author correct for bias in the evaluated groups?
The boys were instructed to avoid vigorous exercise within 2 hours before beginning the test. How was this performed? How does the author control non-exercise play in children between the ages of 10 to 12?
There were any other factors that limited this study? They were taken into account? How was corrected if those factors were used for data analysis?
The author indicates that athletes were trained for 2+ years, this indicated that data correspond to long-term training performance. Does this apply to short-term training? How about regular children? How about other characteristics between participants, other include criteria to reduce the baseline difference? Does the author consider clinical indicators?
Round 2
Reviewer 2 Report
The manuscript has been edited and can be publish in this terms.
Author Response
Thank you very much for your comment and suggestion. You have made our paper better.
Reviewer 3 Report
The authors clarified several of the questions I indicated in my previous review. Most of the questions were addressed on the review clarification paper, but those clarifications were not added to the paper. If added, sentences were added to the end of the paragraphs and in places that don’t mix sense. The author should take a little bit of time and added those clarification in places where it corresponds and explain properly to a broad audience.
Figure 2, hard to differentiate between groups (post-IMT).
Need grammar review. For example, in line 263 “… and nor post-IMT…”
Line 9: “…performance are inconsistent.” … incomplete sentence: to what? Compared to? Should be ‘…performance is inconsistent.’
Lines 59 to 61 need reference, a similar sentence is found in Carey 2007 and LoMauro 2018.
Line 100… H2O correct to H2O
The inclusion criteria should be next to participants, line 81 instead of placed in lines 113 to 115.
